# Beyond Prostate Cancer: An Androgen Receptor Splice Variant Expression in Multiple Malignancies, Non-Cancer Pathologies, and Development

**DOI:** 10.3390/biomedicines11082215

**Published:** 2023-08-07

**Authors:** Kimberley D. Katleba, Paramita M. Ghosh, Maria Mudryj

**Affiliations:** 1Veterans Affairs-Northern California Health Care System, 10535 Hospital Way, Mather, CA 95655, USA; kdkatleba@ucdavis.edu (K.D.K.); paghosh@ucdavis.edu (P.M.G.); 2Department of Medical Microbiology and Immunology, 1 Shields Avenue, UC Davis, Davis, CA 95616, USA; 3Department of Urologic Surgery, 4860 Y Street, UC Davis, Sacramento, CA 95718, USA; 4Department of Biochemistry and Molecular Medicine, 1 Shields Avenue, UC Davis, Davis, CA 95616, USA

**Keywords:** androgen receptor, splice variants, malignancies, pathologies, development

## Abstract

Multiple studies have demonstrated the importance of androgen receptor (AR) splice variants (SVs) in the progression of prostate cancer to the castration-resistant phenotype and their utility as a diagnostic. However, studies on AR expression in non-prostatic malignancies uncovered that AR-SVs are expressed in glioblastoma, breast, salivary, bladder, kidney, and liver cancers, where they have diverse roles in tumorigenesis. AR-SVs also have roles in non-cancer pathologies. In granulosa cells from women with polycystic ovarian syndrome, unique AR-SVs lead to an increase in androgen production. In patients with nonobstructive azoospermia, testicular Sertoli cells exhibit differential expression of AR-SVs, which is associated with impaired spermatogenesis. Moreover, AR-SVs have been identified in normal cells, including blood mononuclear cells, neuronal lipid rafts, and the placenta. The detection and characterization of AR-SVs in mammalian and non-mammalian species argue that AR-SV expression is evolutionarily conserved and that AR-SV-dependent signaling is a fundamental regulatory feature in multiple cellular contexts. These discoveries argue that alternative splicing of the AR transcript is a commonly used mechanism that leads to an expansion in the repertoire of signaling molecules needed in certain tissues. Various malignancies appropriate this mechanism of alternative AR splicing to acquire a proliferative and survival advantage.

## 1. Introduction

The androgen receptor (AR) has a dominant role in mammalian physiology from early development, where it is critical in establishing the male phenotype, to post-natal development and adult maintenance of multiple tissues [1,2,3,4]. While expressed in most cells, this ligand-regulated transcription factor is dependent on androgen (e.g., testosterone or dihydrotestosterone (DHT)) binding for activation [5,6]. Testosterone is secreted by testicular interstitial (Leydig) cells and, at much lower levels, by the adrenal cortex and ovaries [2,7,8]. In addition, testosterone can be converted by 5α reductase to DHT, which binds to the AR with higher affinity [2,7]. There is a ~15-fold higher plasma concentration of testosterone in males than females, resulting in greater AR signaling [9].

Since the AR is a member of the 48-member steroid receptor superfamily, it has the same basic structure as other family members, consisting of an N-terminal activation (NTD), DNA binding (DBD), and ligand binding domains (LBD), where a hinge region lies between the DBD and LBD [3,4,10]. The AR gene maps to the X chromosome and consists of eight exons [11,12]. The NTD is encoded by exon 1, exons 2 and 3 code for the DBD, and the C-terminal LBD is encoded by the terminal four exons. Exon 4 encodes the hinge region, which bridges the DBD and LBD and contains a nuclear localization signal [5]. Prior to ligand binding, the AR is inactive and complexed in the cytoplasm with heat shock proteins (HSPs) [13]. Ligand binding results in a conformation change, release from cytoplasmic HSPs, exposure of the nuclear localization signal, phosphorylation, and ultimately translocation into the nucleus. The dimerized AR DBD tethers the receptor to AR regulatory elements (AREs) on the target DNA to serve as a platform for the assembly of regulatory molecules to drive or repress transcription [6,14,15]. While the LBD and DBD X-ray structures have been solved separately, more recently Yu and collaborators defined the three-dimensional structure of an androgen-occupied AR bound to DNA using cryo-electron microscopy (cryo-EM) [14]. This study confirms that AR dimerizes in a head-to-head and tail-to-tail fashion, shows that the LBD, DBD, and NTDs all have a dimerization interface, and shows that the LBD and the NTD interact. Cryo-EM shows that the NTDs wrap around the LBD and have a major role in recruiting and interacting with SRC-3 and p300 co-regulators (Figure 1).

While the AR has a critical role in male development and physiology, it is a pivotal regulatory molecule in prostate cancer (PCa). Its central function in the initiation and progression of this malignancy was proposed by Huggins and Hodge over 80 years ago [16]. Their visionary studies established that PCa is a hormonally dependent cancer and that limiting AR activity by reducing testosterone levels is efficacious in limiting disease. Hence, for the last eight decades, androgen deprivation therapy (ADT) has been the chief strategy for treating advanced PCa [17,18,19,20,21]. Unfortunately, while initially effective, this therapeutic intervention ultimately fails due to the development of castration-resistant cells [22,23,24]. Multiple mechanisms lead to castration-resistant prostate cancer (CRPC), where the cells remain reliant on AR, but AR signaling is no longer dependent on testosterone [25].

The emergence of AR splice variants (AR-SVs), which retain the NTD and DBD but no longer express the LBD and are therefore refractory to therapies that target this region, is greatly associated with the acquisition of castration resistance. Some AR-SVs retain the ability to translocate into the nucleus, bind to DNA, and regulate gene transcription constitutively [26,27,28,29,30,31].

Below, we summarize our comprehensive review of studies on AR-SV expression and function in multiple non-prostatic malignancies, non-cancer pathologies, and normal tissues in humans and non-human species. This comparative analysis underscores the wide prevalence of AR-SVs in differing cells, where these regulatory molecules make unique contributions to defining cellular physiology in various pathologies and normal processes.

## 2. AR Splice Variants Expressed in Prostate Cancer

### 2.1. Classification and Signaling

Historically, lower molecular weight (LMW) AR variants have been reported in studies on AR in androgen insensitivity syndrome and in malignancies, but their role in PCa was established only about a decade ago [32,33,34,35,36]. While several mechanisms can give rise to these AR forms, including mutations leading to premature termination signals and proteolytic processing that removes the LBD, splice variants (SVs) are the most prominent and best studied [26]. Multiple AR-SVs have been uncovered, and most encode proteins that share common features: they retain the NTD and DBD, are missing the LBD, but acquire a variant-specific short C-terminal sequence encoded by aberrant splicing to a cryptic exon within an intronic region [37]. AR-SVs can be broadly classified into three groups: (1) constitutively active, (2) conditionally active, and (3) inactive [38]. The most extensively characterized variant 7 (AR-V7, aka ARV3) retains the NTD and DBD, while the hinge and LBD are replaced by a 15-amino acid sequence that is derived from intron 3 sequences. Many splice variants are not transcriptionally active, but AR-V7 is representative of constitutively active variants, which can translocate into the nucleus and drive gene expression [5].

In the context of PCa, full-length-AR (FL-AR) and AR-V7 DNA binding site specificities, either homo or heterodimers, have been explored in several studies, but the results are inconsistent; some find that FL-AR and AR-V7 heterodimer and FL-AR homodimer binding sites exhibit significant overlap, while others report a unique AR-V7 binding profile [37,39,40,41] (Figure 2). AR-V7 binding to unique sites is enabled by interactions with co-regulators, including HOXV13 and ZFX, and AR-V7 binding favors open chromatin regions proximal to the transcriptional start site [40,41]. Distinct interactions with co-regulators would be expected since removal of the LBD would significantly alter the three-dimensional structure of a dimerized molecule, either exposing or masking regions capable of interaction with coregulators. Therefore, the FL-AR and AR-V7 transcriptomes have some common targets, but most are unique [33,35,40,42,43]. Multiple studies have established that, in PCa, AR-V7 expression is elevated in CRPC. This led to the development of diagnostics for the detection of AR-V7 in circulating tumor cells [30,44,45,46].

In PCa, expression of AR-SVs has two well-described etiologies: genomic rearrangements of the AR locus and aberrant RNA splicing. Genomic rearrangements associated with AR-SVs have been described in PCa in vitro models and in human CRPC lesions [32,47,48]. The alterations in patient samples are patient-specific, not recurrent, and most specimens harboring genomic alterations express AR-SVs, but there are exceptions [49,50]. AR-SVs are also expressed in cells that have no AR genomic alterations; therefore, a clear correlation between genomic alterations and AR-SVs is not firmly established.

Aberrant RNA splicing is sensitive to FL-AR signaling and therefore can be rapidly initiated in the absence of androgens but reversed when androgen dependent signaling is reinstated, arguing that this process is an adaptation rather than a clonal selection of AR-SV-expressing cells and that androgen deprivation drives AR-SV-expression [33,51,52]. Notably, AR-SV expression is associated with AR gene amplification or transcript elevation; hence, high levels of AR expression may enable AR-SV detection [30,38]. However, other mechanisms may underlie the expression of AR-SVs [53]. The splicing factor hnRNPA1 is upregulated in PCa cell lines harboring AR-SVs, and its recruitment of AR-SV mRNA splice junctions but not FL-AR mRNA splice sites is increased [52]. Splicing factors U2AF65 and ASF/SF2 recognize binding sites near AR exon 3 and facilitate the recruitment of the RNA splicesome to the AR-V7 3′ splice site [54]. Studies of HSP90 have implicated this molecular chaperone in AR-SV splicing, and HSP90 inhibition disrupts this process, resulting in decreased AR-V7 production [55]. The noncoding RNA PCGEM1 and splicing factors also contribute to mRNA splicing, and androgen deprivation induces PCGEM1 redistribution into nuclear speckles and facilitates the interaction of PCGEM1 with U2AF65 [56]. Thus, mechanisms governing splicing are likely cell-context-specific and subject to multiple layers of regulation.

Multiple animal models of cancer have been developed, but surprisingly, there is little information on AR-SVs in these models. The best studies are PCa models developed by genetically engineering mice (reviewed [57]). Many models of aggressive cancer are representative of neuroendocrine malignancies and do not express AR. However, the importance of AR-SVs has been directly addressed by modeling AR-V7 and AR-V^567es^ in transgenic mice where prostate epithelial cell-specific AR-SV forced expression was driven by a modified probasin promoter [58,59]. AR-V7 overexpression promoted the development of prostatic interstitial neoplasia-like lesions. Notably, AR-V7 altered the expression of genes associated with prostate stem cells/progenitor cells and microRNAs deregulated in human cancers. Additionally, the upregulation of TGFβ signaling-associated genes suggested that AR-V7 overexpression may promote epithelial-mesenchymal transition. Prostate-specific expression of AR-V^567es^ drove early hyperplasia and ultimately the development of invasive adenocarcinoma. Transcriptome studies identified enriched expression of inflammatory-related cytokines, transcription factors, and tumorigenesis-associated transcripts. Together, these studies confirm that AR-SV expression reprograms cellular transcription, elevating the expression of genes that promote tumor initiation and progression (Figure 3).

### 2.2. Developmental Therapy Targeting Androgen Receptor Splice Variants

Strides have been made in the development of agents that target the AR-NTD with the goal of inhibiting NTDs containing AR isoforms. Unlike the LBD, the intrinsically disordered AR-NTD makes structure-based drug design difficult, but inhibitor development is underway. In a recent review, Ji and co-authors surveyed the status of NTD-targeting agent development, which can broadly be assigned to four groups [60]:

EPI-001 and the analogues are bisphenol A diglyceride derivatives initially isolated from murine sponge. The original isolated compound was reported to target the AR-NTD [61,62]. While its mechanistic basis has been explored, EPI-001 has a secondary mechanism of action where it was found to selectively modulate peroxisome proliferator-activated receptor-gamma [63]. The most recent derivative, EPI-7386, has a favorable safety, absorption, distribution, and metabolism profile and is able to overcome resistance to current anti-androgen therapies. It therefore exhibits clinical potential and has entered a phase 1 clinical trial (NCT04421222).

Bicalutamide-derived selective AR degraders include several compounds that can directly bind to the AR-NTD of FL-AR and AR-SVs to block interactions with co-activators [64]. The most recently developed degrader demonstrates in vivo efficacy in reducing the growth of castrate-resistant prostate tumors [65].

Agents with dual targeting of STAT3 and AR inhibitory activity have been identified. These compounds interact with the AR-NTD to inhibit recruitment of the AR co-activator SRC-1, resulting in a loss of activity, blocking nuclear translocation of FL-AR and AR-SVs, and causing AR degradation by way of the ubiquitin-mediated pathway [66].

A biphasic antibody molecule that consists of a single-chain variable antibody fragment that recognizes DNA and a single-chain variable fragment that recognizes an epitope in the AR-NTD can accumulate in the nucleus and block signaling by the FL-AR and AR-SVs. This molecule can inhibit the proliferation of AR-dependent cells in a ligand-independent manner [67].

## 3. Sexual Dimorphism of Multiple Cancers

AR signaling has been suggested as a contributing factor in other cancers [68,69]. Varying degrees of male preference are observed in many malignancies, but for several, including the pharynx, larynx, esophagus, and bladder, the incidence in males is over three times greater than that in females (Figure 4). This is accompanied by higher cancer-related deaths in males than females [70]. Since the male and female hormonal milieus differ, where males have higher levels of testosterone while females have greater estrogen levels, the contribution of hormonal signaling, where androgens are tumor promoting and estrogens are protective, has been suggested [71].

Cell culture and animal models reiterate the contributions of androgens and/or AR to the development of cancer. However, clinical studies on the efficacy of AR-targeting therapeutics have been disappointing, showing little measurable benefit. Multiple explanations can reconcile the divergent preclinical and clinical results, including the expression of constitutively active AR-SVs, which, as in CRPC, could render cells refractory to current AR targeting therapeutics.

## 4. Androgen Receptor Splice Variants in Human Malignancies

Below, we review reports of AR-SV expression in non-prostatic malignancies. All the malignancies, with the exception of breast cancer (BrCa), exhibit a sex disparity where males are more likely to develop disease than females. However, counterintuitively, AR-SVs were detected in tumors from males and females, and males did not have an increase in incidence or levels of AR-SVs. In BrCa, a cancer that overwhelmingly occurs in females, AR-SVs are detected in most high-grade malignancies. It is notable that in the context of BrCa, the tumors that most frequently express AR-SVs are Triple Negative BrCa (TNBC), particularly tumors with apocrine features. Moreover, salivary duct carcinomas, the only salivary gland tumor that displays a sex disparity, also exhibit apocrine features and express AR-SVs. While the transcriptomes regulated by distinct AR-SVs have not been investigated in all contexts, the prevailing pattern suggests that the AR-SV-regulated transcriptome is different from the FL-AR-driven transcriptional program. Moreover, there is little, if any, similarity between the investigated AR-V7-regulated transcriptions in different cellular contexts. Lastly, another emerging common theme is that while the level of an SV is low in comparison to the amount of total AR, depletion of the constitutively active AR-SV has a disproportionate effect on the cells’ physiological properties, suggesting that its role is independent of other AR-SVs and FL-AR and needed for optimal cell survival and growth (Table 1).

### 4.1. Breast Cancer

Breast cancers (BrCa) are highly reliant on estrogens and progesterone, whose activity is mediated by their respective receptors [72,73,74]. Hence, in BrCa that expresses these receptors, blocking estrogen receptor (ER) signaling is the first-line treatment strategy. Both the ER and progesterone receptor (PR) are nuclear hormone receptors with structures similar to the AR. The ER is encoded by two distinct genes, ERα and ERβ. These two subunits form hetero or homodimers, which, when bound to estrogens, mediate estrogen signaling. The PR is encoded by a single gene, but two separate promoters are utilized to generate isoforms A and B. AR signaling has been implicated in BrCa as well, but its role in tumorigenesis is less well defined, and epidemiological studies yielded mixed results, even though immunohistochemical studies found that AR is expressed in 31% to 85% of human breast tumors [75,76,77]. Some have identified a significant increase in BrCa risk for women with elevated testosterone and low progesterone levels [78,79]. In other studies, low levels of the androgens dehydroepiandrosterone (DHEA) or dehydroepiandrosterone sulfate (DHEAS) are associated with increased BrCa risk in premenopausal women [80,81,82], while another study did not show a clear association between androgens and BrCa risk [83].

AR as a prognostic marker of disease progression is complicated and nuanced and appears to depend on whether tumors express ER and PR. In ERα-positive malignancies, AR may be a tumor suppressor, while in ERα-negative tumors, AR may be a tumor promoter [84]. AR-regulated genes positively correlate with node invasiveness and independently predict the likelihood of axillary metastases [85,86,87], but AR-positive tumors were associated with better disease-free survival rates [77]. The mechanistic basis of AR-dependent tumor suppression in ERα-positive malignancies may be due to crosstalk between ERα and AR, where the interaction of the two can inhibit the activity of both [88]. AR can compete with ER by binding to ERα response elements or by interacting with co-activators [89]. The AR can also upregulate tumor suppressor genes to directly repress cell growth [90]. Clinical studies that included testosterone for treatment of ERα-positive malignancies either alone or in combination with aromatase inhibitors show that testosterone treatment has clinical benefit [91,92,93].

In contrast, in ERα-negative malignancies, including triple negative (ER-, PR-, and HER2-negative) BrCa (TNBC), AR has oncogenic properties. Blocking AR signaling in ERα-negative cell models suppressed cell growth [94]. The combination of the AR antagonist enzalutamide and HER2-targeting trastuzamb synergizes to inhibit proliferation [95]. However, AR-dependent inhibition of proliferation and induction of cell death may be due to inhibition of WNT/β-catenin and EGFR signaling pathways [96,97]. Significantly, AR has been shown to increase HER2 expression and subsequent activation of MAPK, the WNT/β pathway, and SRC, activating this oncogenic signaling cascade [98,99,100]. Therefore, in ERα-negative malignancies, AR is associated with a poor prognosis [101,102,103], suggesting that AR inactivation therapy may be beneficial [104]. Given these results, AR suppressive agents have been tested either alone or in combination with other drugs in clinical trials in patients with ERα-negative BrCa and have shown efficacy in some patients [105,106,107], but these malignancies are aggressive, and patients who respond quickly relapse [108].

Several studies reported the presence of AR-SVs in BrCa [109,110,111,112,113]. Initial studies reported that five AR-SVs originally identified in PCa (AR-V1, AR-V3, AR-V7, AR-V9, and AR45) are expressed at varying levels in six studied BrCa cell lines. Novel AR-SVs encoding a ninth exon were identified as well [109]. A subsequent study examined AR-SV expression in primary breast tumors. Transcript analysis detected FL-AR in 98.1% and AR-V7 in 53.7% of tumors. AR-V7 protein expression is present in BrCa tumors and in BrCa-derived cell lines. In contrast with what is observed in PCa, FL-AR and AR-SVs are coordinately regulated in BrCa [110]. ERα-negative malignancies have higher levels of FL-AR and AR-V7 than ERα-positive tumors. In a cell culture model, AR-V7 is constitutively active, and activity is not altered by AR antagonists. AR-V7 overexpression in the presence and absence of androgens alters transcription of 64 and 47 transcripts, but only 19% and 11% are upregulated, respectively [110]. 28 transcripts are common to both, and of these, 27 are repressed. Alteration of AR-V7-dependent transcription is distinct from AR-FL, arguing that AR-V7 in this cellular context primarily represses gene expression. Over-expression of AR-V7 modulates transcripts involved in immune function and signaling as well as cell movement. This transcriptome is distinct from what has been observed with AR-V7 overexpression in PCa LNCaP cells, where overexpression primarily alters cell cycle-associated transcripts. These differences are most likely due to distinct cohorts of co-regulators that collaborate with AR-V7 in regulating transcription. Functionally, AR-V7 depletion reduced cell viability, and the reduction was comparable to that of FL-AR, even though AR-V7 is far less abundant. These results further indicate that AR-V7 has distinct pro-proliferative properties in ARα-negative BrCa [110]. Further studies examined the frequency of AR-V7 in BrCa and its correlation with clinical and histopathological features of primary and metastatic malignancies. The majority of AR-V7-positive cases are TNBC, and 71% of these malignancies have apocrine features. In contrast to PCa, AR-V7 is present in primary and metastatic lesions. Given these results, apocrine morphology may serve as a useful screen for the stratification of tumors for the presence of AR-V7 and consideration of AR antagonist therapy [113].

### 4.2. Salivary Duct Carcinoma

Salivary duct carcinoma is a rare cancer of the salivary glands, accounting for 2% of salivary gland tumors. This aggressive malignancy is frequently fatal, and for advanced, unresectable lesions, therapeutic options are limited. Most salivary malignancies do not display a sex disparity, but salivary duct carcinomas are an exception and are more common in males than females [114,115]. Immunohistochemical studies revealed that histologically, these tumors have an apocrine phenotype and share morphological features with invasive ductal carcinoma of the breast [116]. Similar to breast tumors, salivary duct carcinomas stain positive for AR, which is detected in up to 98% of tumors [117,118,119,120]. Notably, several case studies reported clinical benefit from ADT, suggesting that this is a potential therapeutic strategy for patients with recurring lesions [121,122,123,124]. A phase II study of combined androgen blockade (CAB), using leuprorelin acetate and bicalutamide, reported an overall response rate of ~42% of AR-positive malignancies, suggesting that this treatment has promise for recurrent, metastatic, or unresectable locally advanced carcinomas [125]. A larger study of HER2-/AR-positive malignancies found that patients treated with HER2 targeting and ADT therapies showed longer survival [126]. A study on the efficacy of enzalutamide for the treatment of AR-positive malignancies found that while some patients partly responded to treatment, overall, this treatment had limited success [127]. These studies found that while some patients respond to androgen blockade, others that express nuclear AR are refractory to this treatment, highlighting the need for biomarkers that could predict response to androgen deprivation therapies.

AR-SV expression was assessed in a cohort of male and female patients of varying ages [128]. The expression of three different AR-SVs, including AR-V7, was assessed by RT-PCR. AR-V7 is the most frequently detected variant and is present in 37% of tumors. AR-SV expression was confirmed by Western immunoblot analysis. This study also identified AR gene alterations in male and female patients. In vitro studies using salivary duct carcinoma-derived cells showed that depletion of AR using siRNA inhibits cell growth, but cell proliferation is unaltered in steroid-depleted media. The lack of androgen effects was reiterated in an in vivo assay of tumor growth [114]. A study that focused on AR-V7 found that 87% of tumors are positive for AR mRNA, and protein analysis corroborated AR expression. 70% of the AR-positive tumors express AR-V7 mRNA, and AR-V7 protein is detectable in three of five high expressing AR-V7 mRNA samples but is not detected in tumors with lower AR-V7 mRNA levels. In a similar study that evaluated several pathogenic genetic alterations, AR was expressed in all cases, and 50% expressed detectable levels of AR-V7, arguing that AR-V7 is a common feature of these malignancies [129]. Since PCa AR-V7 expression is associated with tumors that are refractory to AR-targeting therapies, AR-V7 expression in salivary duct carcinomas may serve a similar function; hence, detection of AR-SV in salivary duct malignancies may allow for better stratification of patients that would benefit from AR-targeting treatments.

### 4.3. Glioblastoma Multiforme

Glioblastoma multiforme (GBM) exhibits a sex disparity where males are 1.59 times more likely than females to develop this malignancy, and females exhibit a survival advantage [130,131]. Females also exhibit a better response to radiation and chemotherapy [132]. AR protein levels are elevated in 93% of GBMs derived from males and females and from GBM-derived cell lines when compared to normal brain tissue [133,134]. AR expression increases with tumor grade, regardless of sex. Interestingly, 5α reductase 2, an enzyme that converts testosterone to DHT, is elevated in GBMs when compared to normal brain tissue [135]. Glioma cells implanted into nude male and female mice exhibit differential tumor growth; in male animals, tumors are significantly larger and have shorter latency periods, and AR is preferentially detected in tumors propagated in male animals [136]. GBM-derived cell lines expressing AR are sensitive to AR modulation. siRNA-mediated depletion of AR significantly reduces cell proliferation and promotes apoptosis [137]. Likewise, AR antagonist treatments result in decreased cell proliferation, migration, and invasion [138,139,140]. The efficacy of an antagonist was demonstrated in an animal model where treatment with enzalutamide significantly reduced tumor cell growth [140]. Jointly, these studies strongly suggest that AR signaling is pro-tumorigenic in GBM and that AR antagonists curtail tumor cell growth in vitro and in vivo.

AR gene amplification was detected in 27% of GBMs from men and 38% of tumors from women, and further analysis shows that AR mRNA is overexpressed in 93% of tumor samples [140]. This detailed analysis identified a positive correlation between AR expression and age but no correlation between AR levels and survival, gender, race, or ethnicity. AR protein analysis of primary tumor samples and tumor-derived cell lines verified AR expression but also found that 30% of AR-expressing tumors and cells also expressed AR-V7, where the variant is always co-expressed with FL-AR. Treatment of cells expressing both FL-AR and AR-V7 found that they are sensitive to enzalutamide; therefore, in this context, the presence of AR-V7 does not render the cells refractory to this intervention. Thus far, it is unknown if modulating AR-V7 expression in these cells alters the transcriptional output or sensitivity to AR-targeting therapeutics.

### 4.4. Renal Cancer

Epidemiological studies found that males have a two-fold higher risk of developing kidney cancer than females, and this disparity is unaltered by age, year surveyed, or geographic location [141]. While multiple environmental and genetic factors influence the development of renal cell carcinoma (RCC), accumulating evidence supports a role for the androgen signaling axis [142]. Multiple studies report AR expression in RCC [143,144,145,146], but the correlations between AR protein expression and clinical outcomes are conflicting [145,147]. Cell culture and animal models demonstrated the role of AR signaling in RCC. Normal kidney cells transfected with AR-encoding sequences develop larger and larger colonies when treated with carcinogens and promote cell migration and invasion [141]. AR induces vascular endothelial cell proliferation, an effect that is mediated through the PI3K/AKT signaling pathway and results in increased CXCL5 expression [145]. Additional studies reported that the non-coding RNAs HOTAIR and AR are components of a feedback loop that drives transcription of the GLI2 transcription factor, leading to increased expression of the downstream genes VEGFA and PDGFA [147]. Studies of AR-expressing RCC cell lines show that DHT treatment promotes proliferation through STAT5 signaling, and AR inhibition demonstrated antitumor effects [148]. Animal studies utilizing AR-positive RCC tumor cells show that treatment with enzalutamide or limiting DHT production with abiraterone acetate significantly reduces tumor growth [149]. Thus far, there have been no completed clinical trials on the efficacy of hormone deprivation therapy in limiting RCC.

The two most common RCC subtypes are clear-cell RCC (ccRCC) and papillary RCC (pRCC). ccRCC accounts for 75% of all RCCs, while pRCC accounts for approximately 10%. The tumors differ in vascularization; ccRCCs are highly vascularized, while pRCCs are hypovascularized when compared to the surrounding parenchyma [150,151]. ccRCC patients have a worse prognosis than pRCC patients, and approximately 15% of them develop lung, liver, bone, or lymph node metastases [151]. AR splice variant expression (AR-V1, AR-V3, AR-V4, AR-V7, and AR-V12) was investigated using primary ccRCC and pRCC tumors, focusing on AR splice variants that are constitutively active [152]. ccRCC and normal kidney tissues express FL-AR and AR splice variants, but expression is more frequent in pRCC. This result agrees with a previous analysis showing higher AR expression in pRCC than in ccRCC [153]. AR-V7 is the dominant variant expressed, while AR-V12 is not expressed in the analyzed tumors. FL-AR and AR-SV expression are higher in high-grade malignancies (pT4), but there are no significant correlations between patient sex and AR-SV expression. Further inquiries investigated the relationship between relaxin 2 (RLN2), a protein that influences the AR signaling pathway and, in the kidney, is a renoprotective factor whose activity is sex-specific, and AR-SVs [154,155,156]. This study found a significant correlation between FL-AR, AR-V1, and AR-V4 transcript levels and RLN2 expression, a potential modulator of AR activity. More in vitro, in vivo, and genomic studies are needed to define the functional and mechanistic roles of FL-AR and AR-SVs in these malignancies.

### 4.5. Bladder Cancers

Bladder cancer (BlCa) displays a striking sex disparity where males are 3–4 times more likely to develop the disease than females [157]. Numerous studies have reported that the AR has a significant role in BlCa development. Immunohistochemical (IHC) analysis found that AR expression is higher in tumor tissue from males than from females [158]. Moreover, AR staining is more intense and localizes to the nucleus in tumors from males than females [159]. Animal models have confirmed the hypothesis that androgens contribute to bladder carcinogenesis. Supplementation of female rats with testosterone promotes N-butyl-N-(4-hydroxybutyl) nitrosamine (BBN) and induces bladder tumors [160], and urothelium-targeted AR knockout mice have a reduced incidence of BBN-induced tumors [161]. Men with BlCa treated with 5α-reductase inhibitors exhibit a lower risk of BlCa-related death than men not treated with these agents [162]. These studies indicate a significant role for the AR in BlCa. Despite this, two BlCa clinical trials that used the AR inhibitor enzalutamide to target AR were terminated early due to a lack of efficacy.

Most bladder cancer cells express AR-SV, and in some contexts, this is the major isoform expressed [163]. However, the subcellular distribution of AR varies, where some cells have abundant nuclear AR expression while in others the AR is confined to the cytoplasm, and subcellular localization is not dependent on whether the cells were obtained from malignancies in males or females. This study detected AR-V7 expression in ~50% of cells surveyed but also identified a new constitutively active variant, AR-V19, which, like AR-V7, is missing the LBD, contains C-terminal sequences derived from intron 3, localizes to the nucleus, and is constitutively active. Depletion of all AR forms in cells with nuclear AR expression reduces cell viability and increases apoptosis, but AR-V19-specific depletion also effectively reduces cell viability in cells expressing this isoform. Overexpression of FL-AR does not alter AR-SV expression, and conversely, overexpression of V19 does not reduce FL-AR; therefore, the reciprocal negative regulation observed in PCa is not apparent in BlCa. Transcript analysis following depletion of FL-AR, AR-V19, or total AR identified distinct but partly overlapping transcriptomes [164]. mTOR is decreased by deletion of Fl-AR, AR-V19, or all AR isoforms, and this is coupled with altered mTOR downstream signaling. A decrease in mitochondrial components and Increased expression of HIF1α- and HIF1α-regulated transcripts coupled with a significant decrease in oxygen consumption rate on AR depletion argue that an important role of AR signaling is enhanced mitochondrial activity to drive energy production. Taken together, AR signaling in BlCa is appropriated to increase metabolic processes to enhance BlCa proliferation.

### 4.6. Liver Cancer

The AR is implicated in the development and pathogenesis of hepatocellular carcinomas (HCC) [165,166,167]. Mouse HCC models reiterate the contribution of AR signaling, where conditional liver-specific AR knockout decreases the incidence of carcinogen-induced HCC [168]. AR also has a role in Hepatitis B virus (HBV)-induced HCC, an effect that can be attributed to functional AREs in the HBV genome, whereby AR is able to stimulate viral gene transcription, promoting hepatocarcinogenesis [169]. This result is consistent with the observed large differences in male and female incidence of HCC in HBV endemic regions [170,171]. AR protein levels are higher in HCC than in adjacent tissues, and higher AR levels correlate with increased tumor recurrence and decreased overall survival [168,172]. More recent studies find that cytoplasmic AR levels do not differ between tumors and adjacent normal tissue, but nuclear AR expression is significantly higher in tumor cells [173]. Moreover, higher AR mRNA and protein expression correlate with greater overall survival, but higher AR activity, measured as higher expression of androgen-responsive genes, correlates with a higher histological grade and a poorer prognosis [174]. The latter is substantiated by further studies supporting the emerging view that activated and nuclear-localized AR and the corresponding increased AR activity correlate with worse HCC outcomes than overall AR mRNA and protein expression. This suggests that limiting AR signaling would be a viable therapeutic strategy. However, clinical trials of either directly targeting AR with an antagonist or suppressing gonadal androgen synthesis have not shown clinical efficacy [175,176].

One explanation for these disappointing clinical results is that the presence of constitutively active AR-SVs enhances AR signaling and renders the cells refractory to therapeutic interventions. Dauki and collaborators re-examined HCC TCGA patient data and found that 78% of patient tumors expressed AR-SVs, which are detected in male and female patients but are higher in males [177]. This analysis detected the expression of known constitutively active AR SVs, V1, V3, and V7, variants that are active in this cellular context. Depletion of all AR forms reduces cellular proliferation in the presence of androgens, confirming the importance of AR in growth. Overexpression of AR-V7 enhances mTOR signaling, while AR-V7 depletion hinders it, arguing that AR-SVs modulate this critical signaling pathway. Moreover, depletion of AR-V7 reduces cellular invasion and migration, indicative of a role in epithelial mesenchymal transition (EMT). Hence, AR-SVs have varied functions in HCC [177]. It is currently unclear what mechanism(s) drive AR-SV expression, but one explanation is that the observed differential expression of mRNA splicing factors promotes AR-SV expression [178].

## 5. Androgen Receptor Splice Variants in Non-Malignant Cells

In this section, we summarize the studies on AR-SVs in non-cancer conditions as well as in normal tissues (Table 1).

### 5.1. Polycystic Ovarian Syndrome

Polycystic Ovarian Syndrome (PCOS) is the most frequent endocrine disorder of reproductive-aged women, affecting up to 15% of women worldwide [179]. PCOS can cause infertility, hyperandrogenism, and metabolic disorders including obesity, insulin resistance, dyslipidemia, cardiovascular disease, and type 2 diabetes [179]. While the etiology of PCOS has not been completely deciphered, a key feature of this disease is an excess of androgen driven AR signaling [180,181,182]. Excess androgen production is most likely due to increased ovarian output [183]. In the ovary, AR is predominantly expressed by granulosa cells throughout follicular development. Notably, mice with global AR knockout are phenotypically female but have reduced fertility with abnormal ovarian function [184]. Granulosa cell-specific AR knockdown results in premature ovarian failure, decreased fertility, and slower follicle growth. These studies strongly argue that nearly all reproductive phenotypes associated with AR knockdown are due to a lack of AR activity in granulosa cells [185].

Human genetic studies implicate AR CAG repeat polymorphisms and AR SNP rs615G/A with PCOS, suggesting that alterations of the AR gene have a role in PCOS pathogenesis [186,187]. A study of granulosa cells obtained from women undergoing in vitro fertilization and embryo transfer identified two alternatively spliced AR variants expressed exclusively in women with PCOS [188]. One variant is a 69-bp insertion (ins) between exons 2 and 3, while the other has a deletion (del) that skips exon 3; both retain the LBD. Wildtype AR is co-expressed with the splice variants. However, on androgen treatment, the variant AR nuclear translocation is compromised. Notably, women expressing the AR splice variants have higher serum testosterone and DHEA levels than women with PCOS who do not express the variant AR. Chromatin precipitation sequence (ChIP seq) studies following expression of FL-AR or AR-SVs in primary granulosa cells found that the AR-SVs exhibit a different DNA binding pattern than FL-AR. FL-AR binds preferentially to classical AREs, while AR-SVs bind to distinct motifs. Parallel RNA-seq studies found that the FL-AR drives expression of genes related to follliculogenesis, while the AR-SV gene expression profiles differ. Additionally, the expression profiles of the 69 bp ins AR-SV and exon 3 del AR-SV differ from each other as well, indicating that gene cohorts regulated by specific AR-SVs are unique to that AR-SV. In granulosa cells, FL-AR transactivates expression of CYP19A1, the gene encoding aromatase, the rate-limiting enzyme for the conversion of androgens to estrogens, but the AR-SVs are unable to up-regulate this critical enzyme. Therefore, the AR-SV-encoded protein fails to upregulate the ratios of estradiol to total testosterone and estrone to androstenedione. However, the 69-bp insertion in AR-SV activates expression of CYP17A1, which encodes 17α-hydroxylase, which catalyzes androstenedione synthesis. These results argue that granulosa cells harboring AR-SVs have a leading role in the observed excess androgen accumulation in PCOS.

### 5.2. Placenta

The placenta develops at the maternal-fetal interface to support fetal development for nutrient and gas exchange and endocrine signaling. The placenta secretes androgens throughout pregnancy and expresses the AR nuclear receptor and the non-genomic membrane receptor, suggesting a role for androgen signaling in placental function. Androgens and the FL-AR are expressed in the placenta of humans [189,190], rodents [191], sheep [192], and cows [193] for the transcriptional regulation of genes involved in fetal growth and development.

On average, male fetal growth leads to increased birthweights and birthweight centiles compared with female fetal growth in normal pregnancies [194]. Circulating testosterone levels in women with preeclampsia are elevated 2–3 fold, and preeclampsia is associated with elevated placental AR expression. Animal models recapitulated these findings [195]. The importance of understanding placental androgens and AR has been shown during intrauterine development and in a sex-specific manner associated specifically with pregnancy complications. Maternal complications during pregnancy can impact fetal growth in utero, in part due to altered androgen signaling, and alterations in placental AR expression and localization may contribute to disrupted fetal development [196]. Women with preeclampsia have elevated androgen levels and AR gene expression, and it has been hypothesized that androgens affect vascular function and that vascular smooth muscle is involved in the development of preeclampsia. Moreover, it is postulated that an abnormal level of androgens negatively impacts placental angiogenesis and/or alters trophoblast cell proliferation and differentiation [195,197], although the exact mechanisms are unclear. In human pregnancies resulting in preeclampsia, there is an increase in placental FL-AR mRNA and protein expression, regardless of sex [190]. In AR-knockout mice, abnormalities in intrauterine growth and development are more pronounced in male fetuses [198].

AR-SVs have recently been reported in the human placenta which include the FL-AR, AR-V1, AR-V7, and AR-45 along with unknown proteins that are detected by anti-AR antibody including 120, 90, and 55 kDa isoforms [196]. These variants in the human placenta vary by fetal sex, cellular localization, and in the presence of pregnancy complications [196]. AR-FL, AR-V1, AR-V7, and AR-45 have also been recently reported in sheep placenta and their distribution and cellular localization are altered in a pregnancy complication including asthma [199]. In this sheep model for maternal asthma there is a decrease in nuclear expression of AR-45 and cytoplasmic expression of V1, proposing that AR-45, not AR-FL, may be responsible for growth [199]. Understanding placental androgen AR-SV signaling, could give more insight to certain developmental diseases and fetal mortality.

### 5.3. Testis

The AR is essential for the development of testicular tissue [200]. Sertoli cells, the somatic cells present in seminiferous tubules, are subject to AR signaling, which regulates Sertoli cell proliferation and maturation and the maintenance of the blood-testis barrier between cells [201]. Moreover, AR-dependent activities are also important for the surrounding germ cells [201]. Alternative splicing of nuclear hormone receptors is most frequently detected in testicular tissue, and alternatively spliced isoforms of steroid hormone receptors are commonly generated [202]. Several alternatively spliced ER isoforms have been found in the testis, where they modulate ER activity and may have a role in the regulation of spermatogenesis [203,204,205,206,207,208,209].

Early studies on AR-SV expression in testis identified AR45, an AR-SV that encodes a truncated NTD but a complete DBD, hinge, and DBD. Functional studies of this variant in non-testicular tissue found that AR45 could activate or repress AR transcriptional activity, depending on co-regulator expression, but studies were not conducted in the context of testicular cells [210]. Subsequent studies identified novel AR-SVs in human testis that encoded the NTD and deletions/truncations of exons 2, 3, or 4 [211]. ARΔ^23stop^ (a deletion of exon 2, partial retention of intron 2, encoding a protein with only the NTD) and ARΔ4 (in-frame deletion of 120 nucleotides of exon 4 resulting in a protein with an incomplete LBD) were only detected in testis, while ARΔ^2stop^ (premature stop encoding only the NTD) and ARΔ^3^ (in-frame skipping of exon 3 encoding a protein missing the 2nd Zn finger) were only detected in testis. This AR-SV is the same as the AR-SV described in PCOS, in breast cancer samples, and in patients with AIS [211,212,213], which was detected not only in the testis but in other tissues as well.

Studies on impaired spermatogenesis found AR-V7 (called AR3 in the study) is differentially expressed in testicular tissue from controls and patients with nonobstructive azoospermia (NOA) [214]. The NOA patient tissue was further stratified into patients with spermatogenic arrest at the levels of round spermatids (MA), patients with hypo-spermatogenesis (hypo), and patients with Sertoli cell-only syndrome (SCOS). Control tissue expresses FL-AR and AR-SV, while tissues from NOA patients all express similar levels of FL-AR but decreased levels of AR-SV (Control > hypo > MA > SCOS). There is a negative correlation between AR-V7 and serum follicular-stimulating hormone (FSH) levels. Moreover, FSH levels negatively correlate with the Johnsen score, a scoring system that quantifies spermatogenesis according to the profile of cells in seminiferous tubules. Decreased AR-V7 but not Fl-AR levels correlate with NOA, arguing that AR-V7 has a distinct function in spermatogenesis. Future studies will be needed to define the AR-V7 transcriptome and uncover the mechanistic basis of these observations.

### 5.4. Neuronal Lipid Raft

Membrane AR has not been as well studied as classical cytoplasmic/nuclear AR. The AR is expressed in many areas of the brain, including the substantia niagra, hippocampus, and entorhinal cortex [215,216,217]. Some non-DNA-binding-dependent actions of androgens through membrane-bound protein receptors for intracellular signaling have been suggested [218], and cell-impermeable androgens can have fast cellular effects [219,220,221,222,223,224]. Testosterone conjugated to bovine serum albumin leads to rapid membrane effects in N27 and SH-SY5Y neuronal cell lines, resulting in intracellular calcium release [222,223]. In the prostate LNCaP cell line, non-DNA-binding-dependent actions of androgens through membrane AR led to apoptosis and the reduction of tumor size [225]. It has been proposed that this binding antagonizes DNA-binding-dependent actions, arresting androgen action in target tissues.

Recently, Garza-Contreras et al. [226] reported expression of AR45 in a dopaminergic N27 cell line and in brain regions including the substantia nigra pars compacta, entorhinal cortex, and hippocampus from young and middle-aged gonadally intact and gonadectomized rats. The AR45 splice variant that has been characterized in humans lacks the NTD, decreasing the protein molecular weight from 110 kDa to 45 kDa [210]. AR45 is expressed in muscle, lung, heart, breast, uterus, and prostate [210,227,228] and in human brain tissue from an aged population [109]. In dopaminergic neurons, AR45 localizes to plasma membrane lipid rafts and interacts with membrane-associated G proteins Gαq and Gαo, proteins involved in modulating intracellular calcium release via the PLC/IP3 pathway. Hence, the AR-SV may function as a membrane AR to mediate fast, nongenomic androgen actions (Table 1) [226].

## 6. Androgen Receptor Splice Variants in Immune Cells

### 6.1. Peripheral Blood Mononuclear Cells

The immune responses of males and females differ [231]. Females have greater antibody responses to antigens and reject skin allografts more rapidly, hallmarks of greater cellular immune responses that are advantageous for surviving infectious diseases [232,233]. Females have a reduced incidence of tumors but, conversely, an increased incidence of autoimmune diseases consistent with greater cellular immune responses [232]. Sex differences were reproduced in models of autoimmune diseases and led to studies of sex differences in immune response to distinguish between the effects of hormones and genetic factors [234]. Lymphocytes have a central role in autoimmune disease development and progression [232] and express nuclear sex hormone receptors, which control epigenetic and transcriptional processes that can regulate expression of key immune-related genes either directly or indirectly [235,236]. The protective effects of testosterone may underlie the decrease in male susceptibility to autoimmune diseases. In the mouse model of nonobese diabetic disease, castration of males increased disease prevalence, while supplementation of females with testosterone lowered disease incidence [237,238].

These studies led to an evaluation of AR expression in distinct populations of peripheral blood mononuclear cells (PBMC), including B and T lymphocytes, monocytes/macrophages, neutrophil lineage cells, and natural killer cells [231]. Most neutrophils express AR, and androgens can promote neutrophil differentiation and recruitment in humans and mice. In the mouse model, deletion of the AR gene results in severe neutropenia, and neutrophil numbers decrease in mice following castration. AR-deficient animals exhibit normal phagocytic properties but respond less to granulocyte-colony stimulating factor induced proliferation and to migratory signals in vitro. They also exhibited greater susceptibility to apoptotic stimuli and produced fewer proinflammatory chemokines and cytokines. In line with these findings, prostate cancer patients undergoing androgen blockade display neutropenia.

Sub-populations enriched for CD4^+^ T lymphocytes (helper cells), CD8^+^ T lymphocytes (effector cells), and macrophages all express AR, while AR expression in B lymphocytes is much lower [239]. In vitro treatment of CD4^+^ lymphocytes with physiologically relevant levels of DHT results in increased expression of IL-10, a cytokine that suppresses the function of NK cells and T cells while enhancing B-cell survival. This suggests that androgens modify lymphocyte cytokine expression, altering the male immune system [239].

PBMC obtained from non-cancer patients express not only FL-AR but also AR-V7 [229]. Of the PBMC subpopulations studied, AR-V7 is detected in monocytes in all patients and in T- and B-cells and natural killer cells in some patients. FL-AR is detected in all subpopulations. FL-AR and AR-V7 assessment in PBMC and circulating tumor cells (CTC) from PCa patients treated with abiraterone (AA), enzalutamide (E), or taxanes found that high PBMC AR-V7 levels correlate with worse AA/E and better taxane responses. In taxane-treated patients, higher FL-AR expression in PBMC correlates with longer progression-free survival. However, high FL-AR and AR-V7 expression in CTC correlates with shorter progression-free survival and overall survival; therefore, expression in PBMC and CTC has differing predictive roles for therapeutic response. It is also notable that CTC-enriched fractions have detectable levels of the white blood cell marker CD45, indicating that they are contaminated with white blood cells. Testing whole blood cells for AR-V7 expression is premised on the assumption that the primary source of AR-V7 expression is CTC [240]. Identification of AR-V7 expression in lymphocytes suggests that CTC contamination with lymphocytes would contribute to the presence of AR-V7.

### 6.2. Fish Immune Cells during Development

AR protein structure is evolutionarily conserved from fish to humans [241]. Androgens in teleosts, ray-finned fish, are mainly mediated by nuclear AR and found in immune-competent organs of the sea bass, including the head-kidney, liver, and spleen [242], as well as in zebrafish [243]. This suggests that androgens may play a role in immune responses in some fish. The gilthead seabream is a seasonally breeding vertebrate marine fish in the Western Mediterranean. As hermaphrodite teleosts, they are male during at least two reproductive cycles, and each reproductive cycle is divided into four stages, including spermatogenesis, spawning, post-spawning, and resting. It is after the second resting stage that testicular involution occurs, which allows for the sex change [244,245]. Post-spawning, in the testis, there is a massive infiltration of immune cells (acidophilic granulocytes), which are regulated by testicular factors and hormones [244,245,246,247]. Both acidophilic granulocytes and macrophages express AR, with its expression being modulated by androgens [248].

Sanches-Hernandez et al. [230] report a constitutively active AR splice variant that lacks the ligand binding domain, AR∆LBD, in the gilthead seabream testis, head-kidney, and acidophilic granulocytes (equivalent to mammalian neutrophils). This variant has the NTD and DBD but lacks the LBD. Wild-type AR and AR∆LBD show different expression profiles in the testis during the reproductive cycles, suggesting that they play different roles. In addition, the expression ratio in the testis is negatively correlated with testosterone and 11-ketotestosterone throughout the reproductive cycles, while there is a positive correlation between the AR and AR∆LBD expression ratios in the head and kidney. In acidophilic granulocytes, AR and AR∆LBD varied during reproductive cycles and presented a positive correlation with testosterone similar to that in the head and kidney. Bacterial immune stimulation of acidophilic granulocytes regulates alternative splicing of AR, depending on reproductive stage. This suggests a crosstalk between endocrine and immune stimuli in the regulation of AR∆LBD and acidophilic granulocyte function in fish. It is worth noting that the ARΔ^2stop^ and ARΔ^3^ SVs are also present in phylogenetically distinct vertebrate species—rat, African clawed frog, and gilthead seabream, a teleost fish [211].

### 6.3. Role of Androgen Receptor Splice Variants in Immunotherapy

Androgen signaling has implications for the response to immunotherapy. In preclinical studies, melanoma growth following anti-PD-L1 treatment was significantly greater in female animals than in male animals. This effect could be in part explained by inhibition of Treg upon PD-L1 blockade and thus an enhanced immune response, suggesting that androgens present in the male may have a negative impact on responses to anti-PD-L1 therapy [249]. A meta-analysis on checkpoint inhibitor efficacy in treatment for melanoma and non-small cell carcinoma found that overall survival was improved by these therapies for all patients, but the greatest therapeutic benefit was observed in men [250]. In a different meta-analysis, which surveyed checkpoint inhibitor efficacy in patients with various solid metastatic malignancies, significant differences in responses between men and women were not observed [251]. Moreover, where patients received a combination of chemotherapies and checkpoint inhibitors, women had better responses [252], hence the ambiguous data on sex-based responses to checkpoint inhibitors. Currently, the contribution of AR-SVs to checkpoint inhibitor therapies has not been documented.

Most of the studies on androgen signaling blockade and treatment efficacy focus on prostate cancer, where androgen deprivation therapy is a standard treatment for advanced malignancies [231]. Most immunotherapeutic approaches gave disappointing results due to a low tumor mutational burden and a “cold” tumor microenvironment [253]. Single-agent checkpoint inhibitor treatments in PD-L1-positive metastatic prostate cancers achieved positive outcomes in only a small percentage of patients, and in PD-L1-negative malignancies, the outcomes were even worse [254]. It has been noted that treatment with enzalutamide upregulated PD-L1 expression, and androgen deprivation is associated with T-cell infiltration of tumors and increased T-cell response in preclinical models [255,256,257]. Clinical trials of the combination treatment of enzalutamide and checkpoint inhibitors were remarkably effective in over 50% of patients [258,259]. Even though AR-SVs are commonly known to contribute to resistance to anti-androgen therapies, their contribution to or correlation with dual hormonal and immune checkpoint blockage has yet to be uncovered.

Expression of AR-SVs in human neutrophils has not been explored, but given their role in fish development and response to bacterial stimuli, such studies would further our understanding of the endocrine-immune axis in humans.

## 7. Implications, Unresolved Questions, and Conclusions

This comprehensive review of numerous studies that identified AR-SVs in various tissues reveals that AR-SV expression is pervasive across tumor and non-tumor tissues and in diverse species. When analyzed in total, the studies lead to several conclusions: (1) AR-SV expression is conserved through evolution, as evidenced by its presence in fish, frogs, and mammalian species. The AR-SV is similar across species and tissues, where most retain the NTD while the LBD is removed. The exception to this is AR45, which is missing the NTD but retains the DBD and LBD. In most variants, the DBD remains intact, but in some cases, the second ZN finger (encoded by exon 3) may be missing. (2) AR-SVs drive variant-specific transcriptomes that are distinct from those of the FL-AR, but there may be overlapping gene targets. Likewise, the AR-SV binding motifs differ from the FL-AR AREs. Studies in PCOS demonstrated that two different AR-SVs, generated by the insertion and deletion of sequences, drive distinct transcriptional programs; therefore, AR-SV-driven transcriptional programs are specific to a splice variant. Hence, AR-SVs are not simply substituting for FL-AR signaling but complementing, extending, or, in some cases, replacing FL-AR-specific gene expression. (3) AR-SV-driven transcription does not require FL-AR since some cells express only AR-SVs. Moreover, in many cells, a particular AR-SV may constitute a small percentage of total AR present in cells, yet its depletion exerts profound effects on cell proliferation, apoptosis, invasion, and migration, further arguing that AR-SVs have distinct and vital roles that are independent of FL-AR. Furthermore, AR-SVs and FL-ARs can heterodimerize. Is the transcriptional program driven by heterodimers distinct from either homodimer? (4) More recent studies have identified novel constitutively active AR-SVs, but most of the above studies focused on AR-SVs that were originally identified as constitutively active in PCa. It is possible that novel AR-SVs exist but remain undetected, and if true, the prevalence of AR-SVs may be even greater. Malignancies that are not as prevalent or as well studied as those reviewed above may also express these molecules where they may contribute to disease or therapeutic responses, so the prevalence of AR-SVs may be even greater. This requires further study. Moreover, defining the importance of AR-SVs in malignancies may expose a therapeutic opportunity. AR-SVs were detected in PCa cells, which are heavily reliant on AR activity; hence, significant efforts have focused on identifying agents that would target these isoforms along with the FL-AR. Since AR-SVs have a role in promoting the viability and tumorgenicity of multiple malignancies, such agents, when identified, could prove beneficial for the treatment of various malignancies where FL-AR and AR-SVs have a demonstrated role in pathogenesis.

## Figures and Tables

**Figure 1 biomedicines-11-02215-f001:**
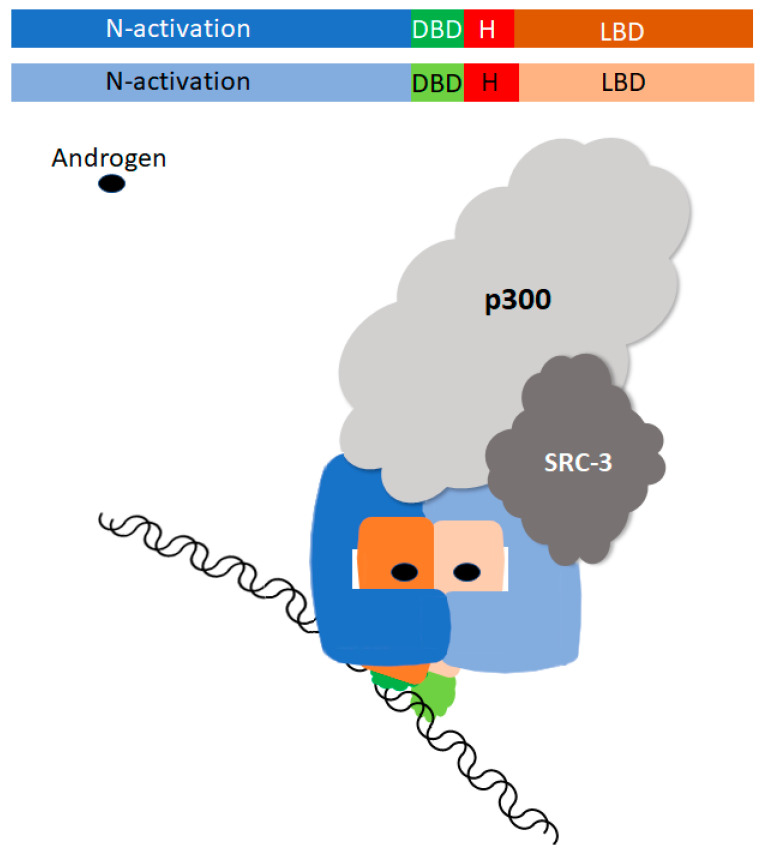
Schematic of the cryo-EM-derived three-dimensional structure of ligand bound AR binding to AREs, interacting with co-regulators p300 and SRC-3. Note that in the dimerized molecule, the NTDs have slightly different confirmations and that the hinge region is hidden. The NTDs interact with each other and wrap around the LBD.

**Figure 2 biomedicines-11-02215-f002:**
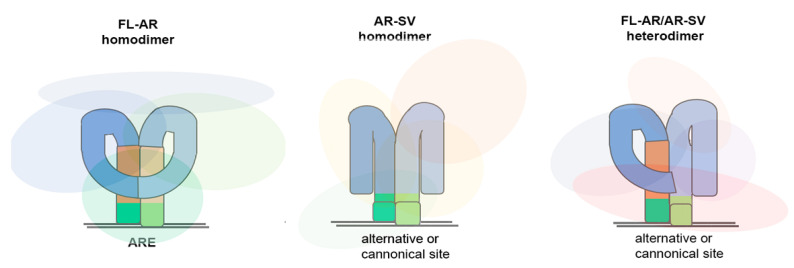
Schematic of homo and hetero dimers of FL-AR and AR-SV. The structures of the AR-SV homodimer and FL-AR and AR-SV heterodimers are unknown; therefore, they are solely the authors’ representation to denote differences. Since the DBD (green) is retained in FL-AR and AR-SV isoforms, the molecules are tethered to DNA sequences. The absence of the LBD (orange) would alter the intermolecular and intramolecular interactions of the NTD (blue); hence, the structure of the dimer would be different, as would the assembly of AR interactors (represented by the light-colored clouds). The unique complement of AR interactors would regulate AR DNA binding, transcriptional activity, and subsequent distinct transcriptional output.

**Figure 3 biomedicines-11-02215-f003:**
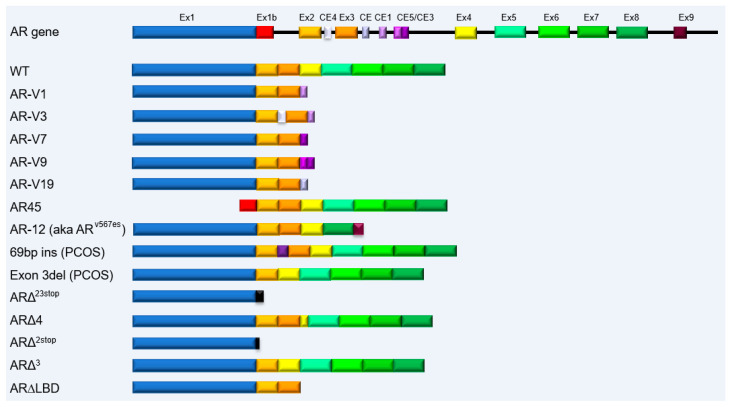
Schematic of the gene encoding AR, noting canonical and cryptic exons that give rise to the FL-AR and a subset of AR-SVs. The exons coding for the different domains are colored NTD-blue, alternative NTD-red, DBD-orange, hinge-yellow, and LBD-green. The cryptic intronic sequences found in AR-SVs are in shades of purple or black.

**Figure 4 biomedicines-11-02215-f004:**
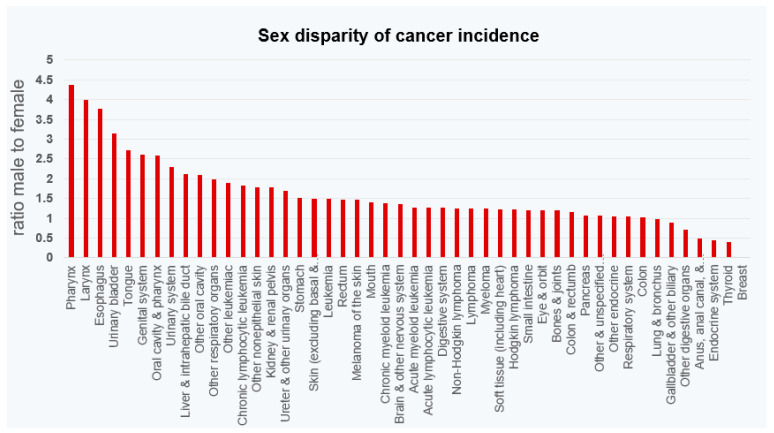
Male-to-female ratio of cancer incidence of the most common cancers (2023) [70].

**Table 1 biomedicines-11-02215-t001:** Summary of AR-SVs detected in various cell types.

	SVs Detected	Biological Processes	Known/Putative Targets	References
**Prostate Cancer**	V1–V14 (and others)	AR signaling, cell cycle, mTORC1, and OX/PHOS	UBE2C, KLK3, EDN2, ETS2, SRD5A1, ORM1 BIRC3, FKBP5, and HES1	[37,39,40,42]
**Breast Cancer**	V1, V3, V7, V45, V9, V15, V16, V17, and V18	immune function, signaling, and cell movement	multiple	[109,110,111,112,113]
**Salivary Duct Carcinoma**	V7	viability/ proliferation	unknown	[128,129]
**Glioblastoma Multiforme**	V7	unknown	unknown	[140]
**Renal Carcinoma**	V1, V3, V4, and V7	expression correlates with RLN2	unknown	[152]
**Bladder Cancer**	V1, V7, and V19	cell cycle, mTOR, OXPHOS, ribosome, mitochondria, and HIF1a	mTOR and FKBP5	[163,164]
**Liver Cancer**	V1, V3, and V7	migration and invasion; mTOR signaling	SNA12	[177]
**Polycystic Ovarian Syndrome**	69 bp ins between exons 2 and 3, exon 3 deletion	limits conversion of androgens to estrogen; fails to upregulate ratios of estradiol to total testosterone	CYP17A1	[188]
**Peripheral Blood Mononuclear Cells**	V7	response to AA/ENZ and taxanes		[229]
**Placenta**	V1, V7 and AR45	unknown	unknown	[196,199]
**Testis**	AR∆2^Stop^, AR∆2^23Stop^, AR∆3, and AR∆4	spermatogenesis	unknown	[211,214]
**Neuronal Lipid Rafts**	AR45	interacts with GPCR proteins Gαq and Gαo	unknown	[226]
**Fish Granulocytes (functional equivalent of mammalian neutrophils)**	AR∆LBD	immunocompetance		[230]

## Data Availability

Not applicable.

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
