# Peer review of "Beyond Prostate Cancer: An Androgen Receptor Splice Variant Expression in Multiple Malignancies, Non-Cancer Pathologies, and Development"

_biomedicines, 2023, doi:10.3390/biomedicines11082215_

Round 1

Reviewer 1 Report

The manuscript deals with the androgen receptor expression and function (with a specific focus on some splicing variants) in neoplastic and non-neoplastic disease.

Overall, the manuscript is well organized and presented. However, the manuscript wants to cover really a lot of topics, looking at several diseases.

This leads to some lacks. For instance, regarding the role of androgen receptors and variants in immune response, I think that the manuscript recently published (PMID 327714315, DOI: 10.3389/fimmu.2020.01184)  considers some points of great interest on the role of androgen receptors and Immune response in tumor immunology and autoimmune diseases. Also, the present manuscript can also update the relevance of immunotherapy (in particular with immune check point blockers) and anti-androgen therapy.

A simple scheme depicting how the androgen receptors functions in some target cells can help the reader to contextualize the topic.

The focus on fish neutrophils can be of interest, but is there no information reported on humans? I understand that the focus of the review is on splice variants, but I would suppose that other manuscripts can be found on other animals regarding the topic of androgen receptors. 

Actually, the review is on humans for some points and on fish for neutrophils, and I would restrict the focus on humans trying to give further details especially on immune response and associated diseases.

The English language is good.

Author Response

Reviewer 1 responses

Thank you for your time and effort reviewing this manuscript. Your insightful questions and comments have been incorporate and the manuscript is better for it.

Q1: This leads to some lacks. For instance, regarding the role of androgen receptors and variants in immune response, I think that the manuscript recently published (PMID 327714315, DOI: 10.3389/fimmu.2020.01184)  considers some points of great interest on the role of androgen receptors and Immune response in tumor immunology and autoimmune diseases. Also, the present manuscript can also update the relevance of immunotherapy (in particular with immune check point blockers) and anti-androgen therapy.

Response: Thanks for this reference and it is included in the manuscript. In the revised manuscript there is an entire section focusing on the issue of AR in immune cells, particularly on the importance of androgen signaling and AR-SV has been expanded incorporating information in the above sited publication. There is a paragraph on AR expression in neutrophils, but we did not find any reports on AR-SVs in these cells, highlighting a gap in knowledge. Additionally, included in this revision is a discussion on the androgen blockade in conjunction with checkpoint inhibitors.

Q2. A simple scheme depicting how the androgen receptors functions in some target cells can help the reader to contextualize the topic.

Response: This was a really good idea. We have included a schematic that is based on the recent cryo- EM derived FL-AR 3D structure. Since AR-SV 3D structure has not been deciphered the pictorial representation is solely the authors’ attempt to differentiate such a structure from that of the FL-AR. AR isoforms missing the LBD would have different configurations and as such would have different exposed (or hidden) surfaces available for interaction with co-regulators. This graphic conceptualizes that the assembled AR-based transcription factor complexes and therefore activities would be isoform specific.   

Q3: The focus on fish neutrophils can be of interest, but is there no information reported on humans? I understand that the focus of the review is on splice variants, but I would suppose that other manuscripts can be found on other animals regarding the topic of androgen receptors. 

Response: Thank-you for this inquiry. We have re-searched the literature and there is no information about AR-SVs in human neutrophils or any mammalian neutrophiles. We have stated this in the manuscript. It is therefore interesting that studies of neutrophils in fish are available while studies in mammals are lacking. Should studies on neutrophiles in human investigate AR-SV, comparative analysis will allow for identification of differences and similarities. Since AR-SV are so common in immune cells, and neutrophiles are very plentiful, this information would further elucidate how AR-SVs regulate expression patterns in immune cells. This review which consolidates AR-SV studies in multiple contexts may promote studies on human or mammalian neutrophils to address this gap in knowledge.

Reviewer 2 Report

This review looks interesting and enough to be published in this journal. However, extensive revision is required.

1. Development status of androgen receptor inhibitors or modulators should be included.

2. Upstream signaling cascade to activate AR and ARSV should be included.

3. Authors have described most of context at the cellular level on ARSV. If authors add some information on ARSV at the organism level like knockout/Tg mice or other organisms, it would be great.

4. References section should be amended according to journal's policy.

Minor check should be required.

Author Response

Reviewer 2 responses

Thank you for your time and effort reviewing this manuscript. Your insightful questions and comments have been incorporate and the manuscript is better for it.

Q1. Development status of androgen receptor inhibitors or modulators should be included.

Response: Thank-you for this suggestion. Since the literature on modulators of FL-AR is vast, particularly in the context of prostate cancer, a section in this review focuses on recent development of agents that target AR-SVs.

Q2. Upstream signaling cascade to activate AR and ARSV should be included.

Response: We have followed this good suggestion and in the current revision we included a paragraph that discusses multiple mechanisms and upstream regulators that lead to the production of AR-SV. This information is from studies in prostate cancer cells, but highlights mechanisms that may be instrumental in other cellular contexts.

Q3. Authors have described most of context at the cellular level on ARSV. If authors add some information on ARSV at the organism level like knockout/Tg mice or other organisms, it would be great.

Response: Thanks for this suggestion. There is surprising little on AR-SVs in animal models, but two studies have directly addressed the role of commonly expressed AR-SVs by generating novel genetically engineered mice where AR-SVs are expressed in prostate epithelial cells. These studies are included in the revised manuscript. 

Q4. References section should be amended according to journal's policy.

Response: We have done this.

Round 2

Reviewer 2 Report

Authors have fully well addressed all issues and therefore it is now acceptable.